# High-pressure crystallography shows noble gas intervention into protein-lipid interaction and suggests a model for anaesthetic action

Igor Melnikov [1,2,11], Philipp Orekhov [3,4], Maksim Rulev[1,2,5], Kirill Kovalev[6,7,12], Roman Astashkin[6,7], Dmitriy Bratanov[1,2], Yury Ryzhykau [7], Taras Balandin[1,2], Sergei Bukhdruker [5,7], Ivan Okhrimenko[7], Valentin Borshchevskiy[7], Gleb Bourenkov[8], Christoph Mueller-Dieckmann[5], Peter van der Linden [9], Philippe Carpentier[5,10], Gordon Leonard [5], Valentin Gordeliy [1,2,6,7✉] & Alexander Popov [5✉]

In this work we examine how small hydrophobic molecules such as inert gases interact with membrane proteins (MPs) at a molecular level. High pressure atmospheres of argon and krypton were used to produce noble gas derivatives of crystals of three well studied MPs (two different proton pumps and a sodium light-driven ion pump). The structures obtained using X-ray crystallography showed that the vast majority of argon and krypton binding sites were located on the outer hydrophobic surface of the MPs – a surface usually accommodating hydrophobic chains of annular lipids (which are known structural and functional determinants for MPs). In conformity with these results, supplementary in silico molecular dynamics (MD) analysis predicted even greater numbers of argon and krypton binding positions on MP surface within the bilayer. These results indicate a potential importance of such interactions, particularly as related to the phenomenon of noble gas-induced anaesthesia.

[1] Institute of Biological Information Processing (IBI-7: Structural Biochemistry), Forschungszentrum Jülich GmbH, Jülich, Germany. [2] JuStruct: Jülich Center for Structural Biology, Forschungszentrum Jülich GmbH, Jülich, Germany. [3] Faculty of Biology, Lomonosov Moscow State University, 119991 Moscow, Russia. [4] Faculty of Biology, Shenzhen MSU-BIT University, 518172 Shenzhen, China. [5] European Synchrotron Radiation Facility, Grenoble, France. [6] Institut de Biologie Structurale (IBS), Université Grenoble Alpes, CNRS, CEA, Grenoble, France. [7] Research Center for Molecular Mechanisms of Aging and Age-related Diseases, Moscow Institute of Physics and Technology, Dolgoprudny, Russia. [8] European Molecular Biology Laboratory (EMBL), Hamburg Outstation, Hamburg, Germany. [9] PSCM (Partnership for Soft Condensed Matter), ESRF, Grenoble, France. [10] Institut de Recherche Interdisciplinaire de Grenoble (IRIG), Laboratoire Chimie et Biologie des Métaux (LCBM), Université Grenoble Alpes, CNRS, CEA, Grenoble, France. [11] Present address: European Synchrotron Radiation Facility, Grenoble, France. [12] Present address: European Molecular Biology Laboratory (EMBL), Hamburg Outstation, Hamburg, Germany. ✉email: valentin.gordeliy@ibs.fr; alexander.popov@esrf.fr

Noble gases are able to interact with protein molecules despite their exceptionally low chemical reactivity. Such interactions have been extensively studied in numerous crystal structure analyses over the past 50 years[1–11]. In these studies, noble gas atoms are shown to bind in various cavities and pockets inside proteins, with their positions easily discernible using anomalous diffraction methods[5]. As these interactions are based on weak London interactions, they could appear to be soft and inconsequential. However, the binding energy of such interactions can easily account for a few kcal/mol (~several kT at 300 K)[12] and it is still far from being clear how substantial the structural effects of noble gas interactions with proteins, including membrane proteins, might be.

One particular attribute of noble gases is their ability to cause anaesthesia. Xenon has been known to be an efficient anaesthetic since the 1950s[13] and has been used clinically for about 30 years. However, the mechanism by which Xe and other anaesthetics cause reversible loss of consciousness is still a matter of debate. Indeed, a number of theories have been proposed to explain the mechanism of general anaesthesia. In the earliest of these, lipids in neuronal membranes were considered as the primary target agents due to a correlation between anaesthetic potencies and lipid/water partition coefficients in a broad range[14] of general anaesthetics—widely known as Meyer-Overton rule[15,16]. In line with this rule, more recent hypotheses suggest that anaesthetics induce changes in physical/chemical properties of the lipidic membranes that modulate the functionality of neuronal membrane proteins[17–22]. However, these theories failed to predict significant effects at clinical anaesthetic working concentrations[23–25]. Nevertheless, the lipidic membrane hypothesis prevailed until the works of Franks and Lieb[23,26–28] who showed inhibition of a protein—a firefly luciferase—in a lipid-free environment by a variety of general anaesthetics, notably at concentrations in good correlation with those required for anaesthesia. A protein alone as the target, however, does not explain why anaesthetic potency of a substance varies in conformity with its solubility in lipids (the Meyer-Overton rule). It would be therefore evident to focus on membrane proteins (MPs).

All research to date suggests that general anaesthetics target neuronal MPs[29]. Among these, such pentameric ligand-gated ion channels[30,31] as NMDA (n-methyl-D-aspartate) receptors[28], two-pore domain K$^+$ channels[32] are modulated by Xe. In contrast, though, GABA$_A$ (gamma-amino-butyric type A) receptors[33,34] show little to no effect upon xenon administration while being a prominent target of other anaesthetics. While structural studies on direct anaesthetic targets are quite limited[35] some surrogate targets like bacterial analogues have provided some insight[31,36] on the molecular mechanism of anaesthetic action. To elucidate a common tendency though, it may be useful to study the effect of anaesthetics on MPs in general.

The other noble gases krypton and argon have also proved to cause anaesthetic effects, but at hyperbaric conditions (minimum alveolar concentration (MAC) in rats being 7.3 bar and 27 bar, respectively[37]), in accordance with their solubilities in fats as was predicted by Meyer and Overton. No anaesthetic effect was observed for lighter neon and helium[37], presumably because these require pressure values[38] far beyond the limits of a nervous system, at which the hyper excitability already takes place. So far, structural studies on MPs with noble gases are rare (as opposed to the studies on soluble proteins) and mainly focused on exploring cavities and substrate pathways inside channel/transporter MPs[39–42]. However, few relevant MP studies have already demonstrated that xenon can modulate MP activity[11] and change its conformational states[31]. All the studies with a minor exception[11] (with only one xenon atom bound) were carried out on lipid-free systems (detergent-mediated type I crystals), hence,

the interaction between noble gases and MPs may not have been fully elucidated due to the absence of the lipid bilayer in which MPs are embedded. It is recognised that lipids play a crucial role in MP functioning by altering its structure[43]. However, there is no solid explanation about the mechanism of such modulation, and structures with alternative arrangements of lipid chains are precious and in demand. The aims of this work therefore are to provide a more genuine insight of MP-noble gas interaction that better addresses the influence of lipids and to provide general insights into the phenomenon of noble gas-induced anaesthesia. For these purposes, we used pressurised argon and krypton atmospheres to produce derivatives of *in meso* grown crystals of three well-characterised MPs. Crystal structure analysis exploiting anomalous scattering techniques was then used to identify noble gas atom binding sites. In complementary studies, we also carried out molecular dynamics (MD) simulations of noble gas atoms binding to single molecules of each MP embedded in a model lipid bilayer and assessed the validity of this method by comparing the results with the crystal structures reported here. Taken together, the results obtained show that, in addition to binding in internal cavities within the MPs, noble gas atoms interact with MP hydrophobic surface at many sites and displace lipid molecules bound to their surfaces. We hypothesise that latter phenomenon may affect MP function, thus providing an insight to the anaesthetic effects of noble gas atoms.

## Results

In this work, we prepared several crystals of a triple mutant of bacteriorhodopsin[44] (tmBR), proton-pumping rhodopsin MAR[45] and sodium pump KR2[46] (monomeric form); these rhodopsins, despite sharing a similar fold, are of different origins and, therefore, are naturally surrounded by different lipids (which makes the MPs quite different in the landscapes of hydrophobic surfaces).

All crystals were derivatised with krypton under the pressure of 130 ± 5 bar using the "soak-and-freeze" methodology at the ESRF high-pressure facility for MX[47]; additionally, tmBR crystals were also derivatised with argon under 2000 ± 10 bar using the high-pressure freezing method developed at the facility[48]. Native datasets were also collected from crystals of the same batch for tmBR and MAR at ambient pressure (Table 1). Krypton/argon atoms (Fig. 1) were located using anomalous difference Fourier electron density maps (see Methods). Datasets of the highest resolution for each derivative were then chosen for refinement (Table 1). In contrast to previously reported studies[31,39–42,49], most of the noble gas atoms were observed on the outer hydrophobic surface (buried in the membrane) of the MPs and only a small portion of the atoms were identified in internal cavities. An explicit correlation can be established between the number of noble gas positions found in the structures and the resolution of diffraction data.

In total, 47, 35, 11 and 19 noble gas atoms were identified as partially occupying positions in the four crystal structures (tmBR-argon, tmBR-krypton, KR2-krypton, MAR-krypton, respectively, see Table 2, Fig. 1). These binding sites can be sorted into three groups: group 1 comprises high occupancy positions in cavities internal to the 7-helices volume of the individual MPs (see Supplementary Note 1, Supplementary Fig. 1); groups 2 and 3 contain positions on their hydrophobic surfaces. The latter are divided into positions exposed to the bulk lipid bilayer region (group 2) and those in membrane plane crystal contacts between the molecules (group 3). The majority of the binding sites belong to group 3. However, the occupancy of atoms enclosed in crystal contacts is approximately the same on average as of those on the surface (group 2) (see Table 2, Methods). A good consistency is observed between argon and krypton binding positions: of 35

**Table 1 Data collection statistics (unmerged Friedel's pairs).**

| Protein acronym | KR2 | MAR | | | tmBR | | | | |
|---|---|---|---|---|---|---|---|---|---|
| Derivative / Dataset type | Krypton High-resolution | Native (2 datasets merged) | Krypton High-resolution | Merged anomalous | Native | Krypton High-resolution | Merged anomalous | Argon High-resolution | Anomalous |
| X-ray wavelength (Å) | 0.8637 | | 0.861 | - | | 0.8634 | - | 0.97242 | 1.85 |
| Space group | I222 | P21 | | | C222 | | | | |
| a (Å) | 40.5 | 50.3 | 50.9 | 50.9 | 115.9 | 115.3 | 115.4 | 115.7 | 115.5 |
| b (Å) | 82.2 | 40.0 | 40.38 | 40.31 | 119.4 | 120.0 | 119.8 | 119.2 | 119.2 |
| c (Å) | 233.4 | 60.2 | 60.62 | 60.81 | 36.2 | 36.5 | 36.4 | 36.4 | 36.3 |
| β ° | - | 102.08 | 101.36 | 101.30 | - | - | - | - | - |
| Resolution (Å) | 100-2.6 (2.8-2.6) | 100-1.85 (1.90-1.85) | 100-2.25 (2.35-2.25) | 100-2.25 (2.35-2.25) | 100-1.70 (1.74-1.70) | 100-2.0 (2.1-2.0) | 100-2.0 (2.1-2.0) | 100-1.65 (1.70-1.65) | 100-2.2 (2.3-2.2) |
| Multiplicity | 6.9 (7.0) | 3.2 (1.9) | 3.5 (3.5) | 9.3 (5.1) | 3.5 (3.5) | 6.7 (6.7) | 15.2 (10.2) | 3.1 (3.1) | 6.3 (5.3) |
| Completeness (%) | 99.9 (100.0) | 98.7 (95.6) | 99.4 (97.9) | 99.5 (96.8) | 99.4 (99.6) | 99.7 (99.8) | 100 (100) | 99.0 (99.4) | 99.2 (96.7) |
| $R_{meas}$ (%) | 19.2 (211.7) | 11.7 (101.3) | 14.0 (129.9) | 21.7 (139.2) | 8.8 (95.7) | 14.9 (186.0) | 21.2 (185.0) | 5.0 (176.9) | 3.3 (48.1) |
| $<I/\sigma(I)>$ | 9.22 (1.07) | 8.37 (1.13) | 7.74 (114) | 9.42 (1.19) | 10.52 (1.33) | 9.09 (1.49) | 11.09 (1.63) | 16.29 (0.92) | 37.72 (4.52) |
| $CC_{1/2}$ (%) | 99.7 (40.0) | 99.6 (45.5) | 99.6 (49.1) | 99.6 (45.0) | 99.8 (59.1) | 99.8 (51.7) | 99.8 (62.8) | 99.9 (34.0) | 100 (95.9) |
| XDS SigAno (100-10 Å shell) | 1.515 | - | 1.352 | 1.449 | - | 2.213 | 2.881 | 1.18 | 4.454 |
| $CC_{ano}$ (%) | 61 | - | 54 | 62 | - | 83 | 85 | 45 | 93 |

*(a) Data collection and refinement statistics (molecular replacement)*

| | KR2 + krypton (high-res dataset) | MAR native | MAR + krypton (high-res dataset) | MAR + krypton (merged anomalous) | tmBR native | tmBR +krypton (high-res dataset) | tmBR +krypton (merged anomalous) | tmBR+argon (high-res dataset) | tmBR+argon (anomalous) |
|---|---|---|---|---|---|---|---|---|---|
| **Data collection** | | | | | | | | | |
| Space group | I222 | P21 | | | C222 | | | | |
| Cell dimensions | | | | | | | | | |
| a, b, c (Å) | 40.5 82.2 233.4 | 50.3 40.0 60.2 | 50.9 40.38 60.62 | 50.9 40.31 60.81 | 115.9 119.4 36.2 | 115.3 120.0 36.5 | 115.4 119.8 36.4 | 115.7 119.2 36.4 | 115.5 119.2 36.3 |
| α, β, γ (°) | | 101.30 | 101.36 | 102.08 | | | | | |
| Resolution (Å) | 100-2.6 (2.8-2.6) | 100-1.85 (1.90-1.85) | 100-2.25 (2.35-2.25) | 100-2.25 (2.35-2.25) | 100-1.70 (1.74-1.70) | 100-2.0 (2.1-2.0) | 100-2.0 (2.1-2.0) | 100-1.65 (1.70-1.65) | 100-2.2 (2.3-2.2) |
| $R_{meas}$ | 19.2 (211.7) | 11.7 (101.3) | 14.0 (129.9) | 21.7 (139.2) | 8.8 (95.7) | 14.9 (186.0) | 21.2 (185.0) | 5.0 (176.9) | 3.3 (48.1) |
| $I / \sigma I$ | 9.22 (1.07) | 8.37 (1.13) | 7.74 (114) | 9.42 (1.19) | 10.52 (1.33) | 9.09 (1.49) | 11.09 (1.63) | 16.29 (0.92) | 37.72 (4.52) |
| Completeness (%) | 99.9 (100.0) | 98.7 (95.6) | 99.4 (97.9) | 99.5 (96.8) | 99.4 (99.6) | 99.7 (99.8) | 100 (100) | 99.0 (99.4) | 99.2 (96.7) |
| Redundancy | 6.9 (7.0) | 3.2 (1.9) | 3.5 (3.5) | 9.3 (5.1) | 3.5 (3.5) | 6.7 (6.7) | 15.2 (10.2) | 3.1 (3.1) | 6.3 (5.3) |

Number of xtals for each structure should be noted in footnote. Values in parentheses are for highest resolution shell.

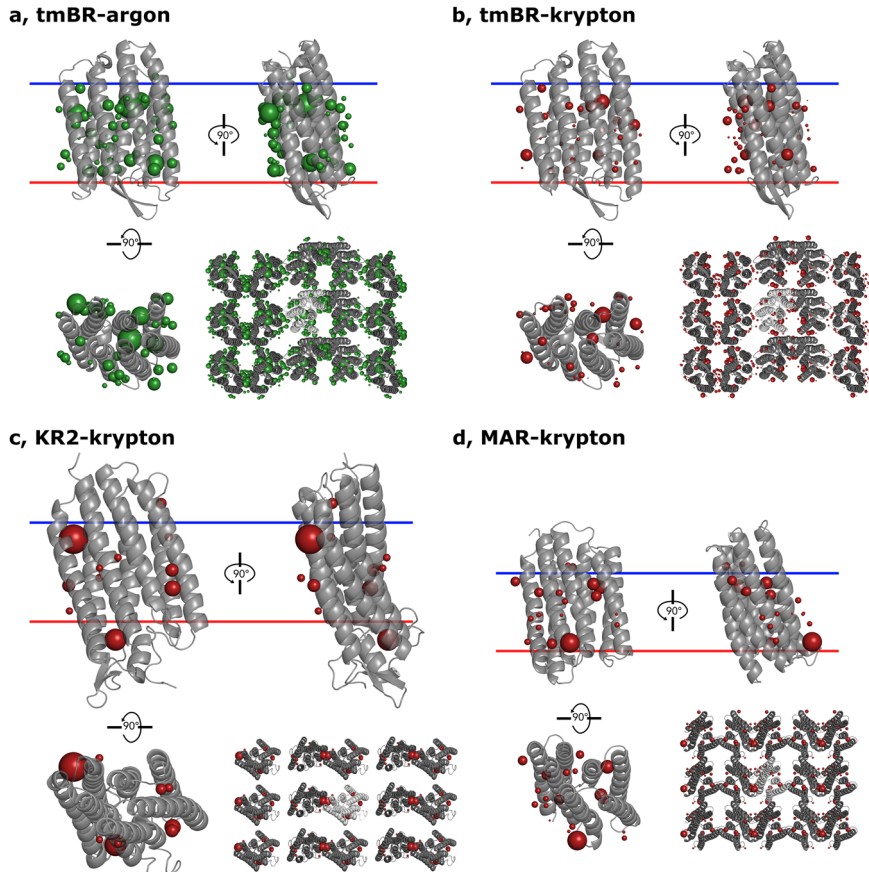

**Fig. 1 Cartoon representations of the structures of tmBR-argon, tmBR-krypton, KR2-krypton, MAR-krypton. a** tmBR-argon. **b** tmBR-krypton. **c** KR2-krypton. **d** MAR-krypton. Each panel shows side views of the proteins sitting in the membrane (top left and top right figures), a top view perpendicular to the membrane (bottom left), and a view of crystal packing contacts in the *XY*-plane (bottom right figure). Noble gas atoms are shown as red (krypton) and green (argon) spheres. Sphere sizes are proportional to atom occupancy.

noble gas atom positions identified in tmBR-krypton, only two—which have low occupancy (0.1)—do not match corresponding sites in tmBR-argon. In general, noble gas atoms prefer interacting with hydrophobic residues (Supplementary Fig. 2) which is not surprising; however, noble gas binding is usually a cumulative interaction of all the residues in a pocket, and, therefore, lacks a certain degree of specificity.

Most of the surface noble gas atom positions are close to the positions of lipid acyl chain fragments seen in the native crystal structures (see Fig. 2, in orange). Moreover, at many positions (11, 13, 8 and 6, for tmBR-argon, tmBR-krypton, MAR-krypton and KR2-krypton, respectively), noble gas atoms replaced, at least partially, lipid atoms, with an altered conformation of lipid chains being sometimes clearly visible (see Fig. 2a). However, in most of the cases this second conformation of the lipid chain was not clear. The other visible lipid fragments were observed at the same positions they occupy in the native structures. Overall, the derivatised structures do not differ significantly from the native ones with $C_\alpha$ r.m.s.d. value not exceeding 0.3 Å (see Table 2, Methods); rare side chain position alteration was observed (Supplementary Figure 3). However, in the particular case of KR2-krypton the backbone of helix E is displaced about 1 Å near the site of two krypton atoms bound in a pocket proximal to ß-ionone ring of the retinal molecule which maintains an unchanged conformation compared to the native crystal structure (see Fig. 3).

To complement our crystallographic studies, we also analysed the probability density maps, obtained by molecular dynamics (MD) simulations (MD maps, see "Methods", Table 3, Supplementary

Fig. 4), of krypton and argon interactions with the proteins studied here embedded into a lipid bilayer. The MD simulations were carried out on monomeric molecular models of tmBR, KR2, and MAR in the presence of excessive concentration of noble gases (see Table 3) sufficiently high to improve the sampling but at the same time causing no aggregation. The result suggested that both noble gas atoms massively populate the hydrophobic surface of the proteins (Fig. 4, Supplementary Videos 1-4). The binding free energy corresponding to these peaks was estimated down to −16 to −19 kJ/mol using the Boltzmann relation[50]. Interestingly, the inspection of individual binding events at some representative sites revealed that noble gases form considerably long-lived contacts lasting for 10–100 ns (see Supplementary Fig. 5). The peaks in the MD maps were systematically clustered (see Fig. 4) suggesting that the atoms may frequent between close positions.

Inspection of the correspondence between noble gas positions in the crystallographic models and the MD simulations shows that almost all noble gas atoms positions observed in the former have a nearby counterpart peak in the latter (see Fig. 4). However, not surprisingly, the positions in crystal contacts and the internal positions were not entirely reproduced by the MD (Table 4) since the simulation was carried out on single molecules and also had a limitation in observation time. On the other hand, noble gas positions "on the surface", which are of concern here, coincided properly with the MD map peaks (Table 4, Supplementary Videos 1–4). In total, the MD experiment predicted obviously a greater number of noble gas positions than those observed in the crystallographic models (see Table 4).

**Table 2 Refinement statistics.**

| Protein acronym | KR2 | MAR | | tmBR | | |
|---|---|---|---|---|---|---|
| Derivative | Krypton | Krypton | Native | Krypton | Argon | Native |
| Resolution limit (Å) | 2.6 | 2.25 | 1.85 | 2.0 | 1.65 | 1.70 |
| $R_{work}/R_{free}$ (%) | 21.6/26.9 | 17.8/24.5 | 16.6/21.2 | 17.0/23.2 | 16.3/20.5 | 15.4/19.5 |
| Number in the ASU: | | | | | | |
| Protein residues | 272 | 217 | 217 | 225 | 228 | 225 |
| Atoms | 2444 | 1983 | 2024 | 2095 | 2240 | 2131 |
| Water molecules | 81 | 99 | 137 | 106 | 119 | 110 |
| Atoms of lipids | 194 | 162 | 177 | 206 | 219 | 243 |
| Other ions | Na$^+$ | – | – | SO$_4^{2-}$ | 2 SO$_4^{2-}$ | 2 SO$_4^{2-}$ |
| Noble gas atoms: | 11 | 19 | – | 35 | 47 | – |
| Internal | 0 | 6 | – | 2 | 5 | – |
| Surface | 7 | 8 | – | 11 | 15 | – |
| In crystal contacts | 4 | 5 | – | 22 | 27 | – |
| R.m.s.d. with native C$_\alpha$ positions (Å) | 0.31 | 0.26 | – | 0.11 | 0.11 | – |
| Average B-factors (Å$^2$) | | | | | | |
| Overall | 46.8 | 34.6 | 34.3 | 29.4 | 27.5 | 22.5 |
| Protein atoms | 51.8 | 35.3 | 33.6 | 28.1 | 25.5 | 19.2 |
| Water molecules | 61.5 | 51.5 | 54.6 | 52.2 | 44.4 | 38.8 |
| Atoms of lipids | 51.2 | 34.6 | 34.8 | 31.8 | 30.2 | 22.5 |
| Noble gas atoms (internal) | – | 32.9 | – | 23.6 | 22.3 | – |
| Noble gas atoms (surface) | 48.5 | 33.8 | – | 25.0 | 21.7 | – |
| Noble gas atoms (crystal contacts) | 44.4 | 40.5 | – | 28.4 | 22.6 | – |
| Average occupancy | | | | | | |
| Lipid atoms | 0.59 | 0.46 | 0.39 | 0.45 | 0.40 | 0.35 |
| Noble gas atoms (internal) | – | 0.30 | – | 0.25 | 0.50 | – |
| Noble gas atoms (surface) | 0.32 | 0.22 | – | 0.22 | 0.29 | – |
| Noble gas atoms (crystal contacts) | 0.38 | 0.26 | – | 0.16 | 0.30 | – |
| Ramachandran plot (%) | | | | | | |
| Preferred | 96.7 | 99.1 | 99.5 | 99.1 | 98.5 | 99.1 |
| Allowed | 3.3 | 0.9 | 0.5 | 0.9 | 1.5 | 0.9 |
| Outliers | 0 | 0 | 0 | 0 | 0 | 0 |

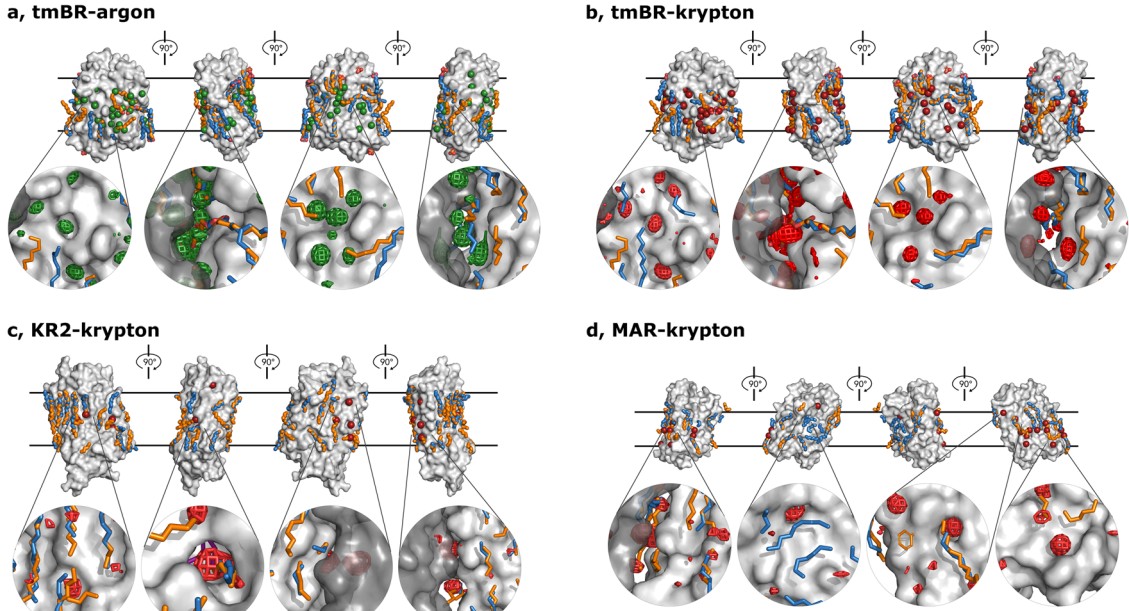

**Fig. 2 Illustration of noble gas positions on the surfaces of the derivatised structures.** tmBR (**a**, argon; **b**, krypton), KR2-krypton (**c**), MAR-krypton (**d**). Each panel shows four orthogonal side views in the membrane. The hydrophobic membrane region is delimited with two black parallel lines. Krypton and argon atoms are shown as red and green spheres, respectively. Lipid fragments present in the derivatised structures are shown in sky-blue; lipid fragments from native structures are shown in orange for comparison. On the bottom of each panel four representative noble gas binding sites in each view are enlarged and the anomalous difference density krypton (red grid) or argon (green grid) are drawn at the 3.0 r.m.s. level. Symmetry-related molecules are shown in transparent dark to better illustrate the contacts.

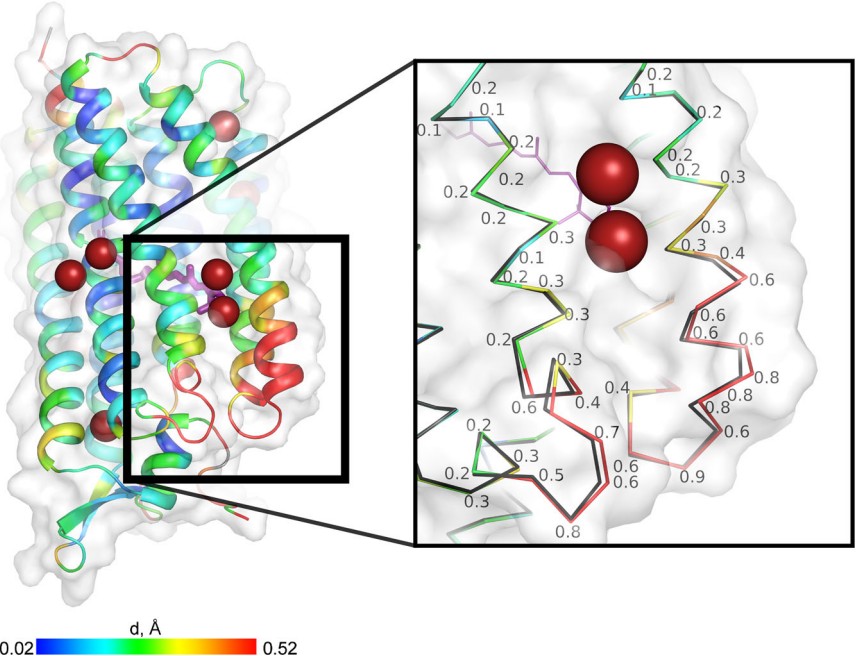

**Fig. 3 Cartoon representation of the crystal structure of KR2-krypton.** Cartoon colour illustrates Cα-position discrepancy (Å) between the derivatised and the native structures (with the scale at the bottom). In the zoom window, those discrepancies are detailed for each atom. Krypton atoms are shown as red spheres.

**Table 3 Details of the simulated systems.**

| System | KR2 + Kr | MAR + Kr | tmBR + Kr | tmBR + Ar |
|---|---|---|---|---|
| Lipid type and number | 125 DPPC | 90 DLPC | 103 DPPC | 103 DPPC |
| Simulation box size, Å | 69.3 × 69.3 × 128.4 | 62.2 × 62.2 × 99.9 | 65.3 × 65.3 × 103.3 | 65.3 × 65.3 × 103.3 |
| Number of noble gas atoms | 240 Kr | 126 Kr | 140 Kr | 140 Ar |
| Total number of atoms | 60211 | 34382 | 40720 | 40720 |
| Probability density map sampling box size, Å | 58 × 43 × 78 | 55 × 45 × 62 | 54 × 45 × 65 | 53 × 42 × 64 |
| Probability density map sampling interval, Å | 0.5 | 0.5 | 0.5 | 0.5 |
| Map mean value, atoms/(0.5 Å)³ | 0.00047 | 0.00046 | 0.00047 | 0.00043 |
| Map standard deviation value, atoms/(0.5 Å)³ | 0.00094 | 0.00095 | 0.00087 | 0.00078 |

Finally, we utilised a simplified computational approach based on the normal mode analysis (NMA) of the coarse-grained elastic network model (ENM) to estimate the change in protein dynamics induced by binding of noble gases. For all the investigated protein systems, we observed slight decrease in dynamics in presence of a noble gas (see Fig. 5). This effect was more prominent for both tmBR structures, in the presence of argon and krypton. A constant negative difference in fluctuations is seen along the whole protein sequence. In fact, that shows the low level of specificity in noble gas action as dynamics suppression is uniform in the structure. Peaks in fluctuation difference come from flexible loops between α-helices—since they are more dynamic in general and the difference is larger. In order to cross-validate the results of NMA, we also conducted a series of short MD simulations with and without the crystallographically resolved atoms of noble gases. The results of these simulations appear in agreement with NMA indicating slightly reduced dynamics of proteins in the presence of argon and krypton (see Fig. 5).

## Discussion

Limitations, including diffraction data resolution, the use of detergent-based crystal systems, and relatively low pressures in previous work[11,31,39,40,49] on MP derivatisation with noble gases,

notably with xenon, limited the number of noble gas atom positions identified and did not allow an extensive probing of their interactions with MPs. In the experiments described here, we extend our knowledge as to how noble gas atoms bind to MP molecules, showing that they readily interact with the outer hydrophobic surface of MPs by non-specific binding at multiple sites and in particular compete with binding of lipids to the surfaces of the systems studied.

In addition to results obtained using macromolecular crystallography (MX), we performed MD simulations in order to validate whether such an in silico method can generate reliable information about binding positions of noble gas atoms at the surface of MPs.

When comparing the results of MX experiments and MD simulations, two conclusions can be drawn. Firstly, noble gas binding sites appearing to involve crystal contacts between MPs (3rd group) are poorly matched with map peaks in the MD simulations. A clear reason for this is that in the latter only a single MP molecule was considered (i.e. crystal contacts do not exist). Indeed, where MX sites in this group are also predicted in MD simulations crystal contacts cannot be the defining characteristic, suggesting that such sites will always be accessible. Secondly, and most strikingly, all the crystallographically observed positions belonging to the 2nd group ("MP surface"

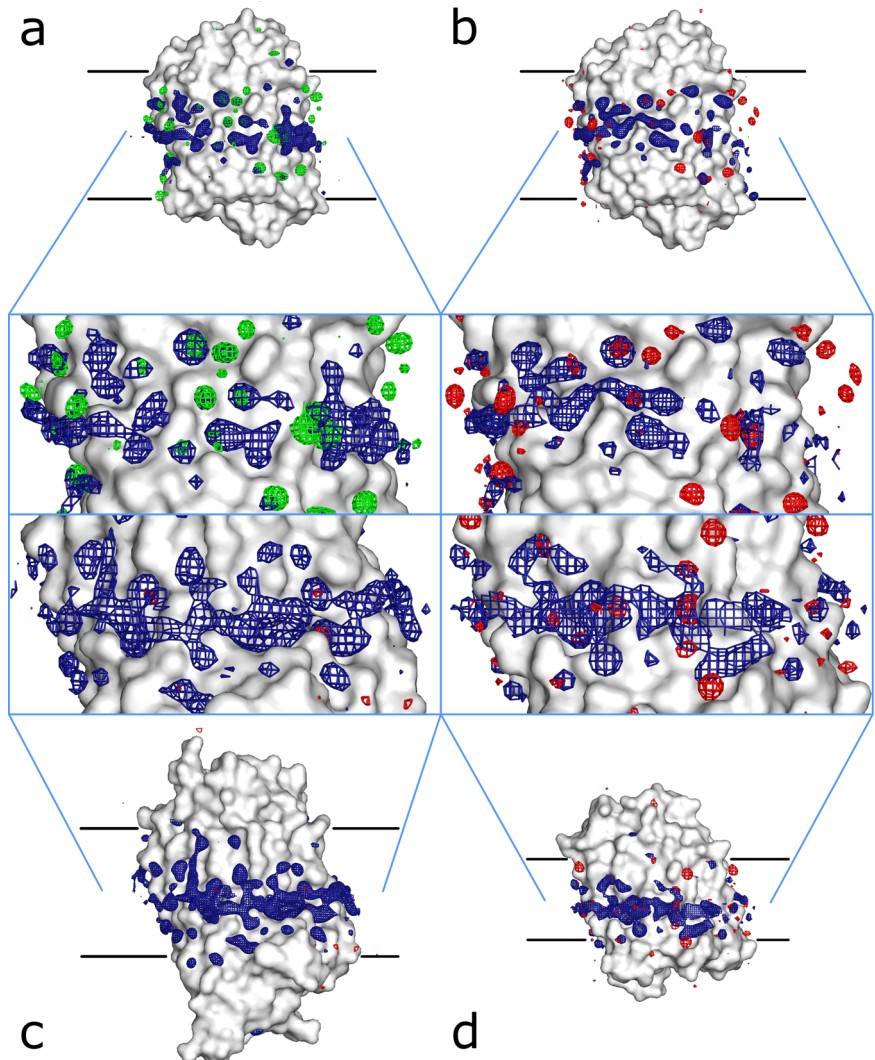

**Fig. 4 Arrangement of peaks of the MD maps.** MD maps are sampled at the level of {mean + 3.5×std.dev.} (navy blue grid) on the surface of the proteins. **a** tmBR-argon. **b** tmBR-krypton. **c** KR2-krypton. **d** MAR-krypton. Peaks in the anomalous difference maps obtained in crystallographic studies are shown (sampled at 3.5×r.m.s. level) in red (krypton) and green (argon).

**Table 4 Correspondence between the MD results and MX models.**

|  | KR2 + Kr | MAR + Kr | tmBR + Kr | tmBR + Ar |
|---|---|---|---|---|
| Noble gas atoms | 11 | 19 | 35 | 47 |
| Matches | 10 | 16 | 30 | 38 |
| "Surface" atom matches, MD/MX | 6/7 | 8/8 | 11/11 | 15/15 |
| "Crystal" atom matches, MD/MX | 4/4 | 5/5 | 18/22 | 23/27 |
| "Internal" atom matches, MD/MX | – | 3/6 | 1/2 | 0/5 |
| Number of MD density peaks | 133 | 96 | 96 | 102 |
| Number of MD density peak clusters | 46 | 37 | 39 | 38 |
| Clusters not present in the MX model | 38 | 31 | 26 | 23 |

positions) were entirely represented in the results of MD simulations except one krypton position hidden in an excavation, passage to which is blocked by histidine side chain in KR2 structure (in fact, this position is not accessible in the MD experiment as are the internal positions, binding to which requires more significant rearrangements in macromolecule structure) (Supplementary Fig. 6). Several additional positions were observed in the MD simulations, which are absent in the crystallographic models. That they are not observed in MX

experiments might be due to insufficient data resolution and/or very low occupancy. These MD-predicted positions without crystallographic counterparts are thus likely to be real and should be considered as such.

Noble gases interact with proteins via weak Van der Waals forces[51]. Mostly, London interactions (spontaneously induced dipoles)[5] occur, and binding sites are thus essentially of hydrophobic nature (apolar pockets). Polarisability is the main factor in such interactions and it drops rapidly from xenon to helium

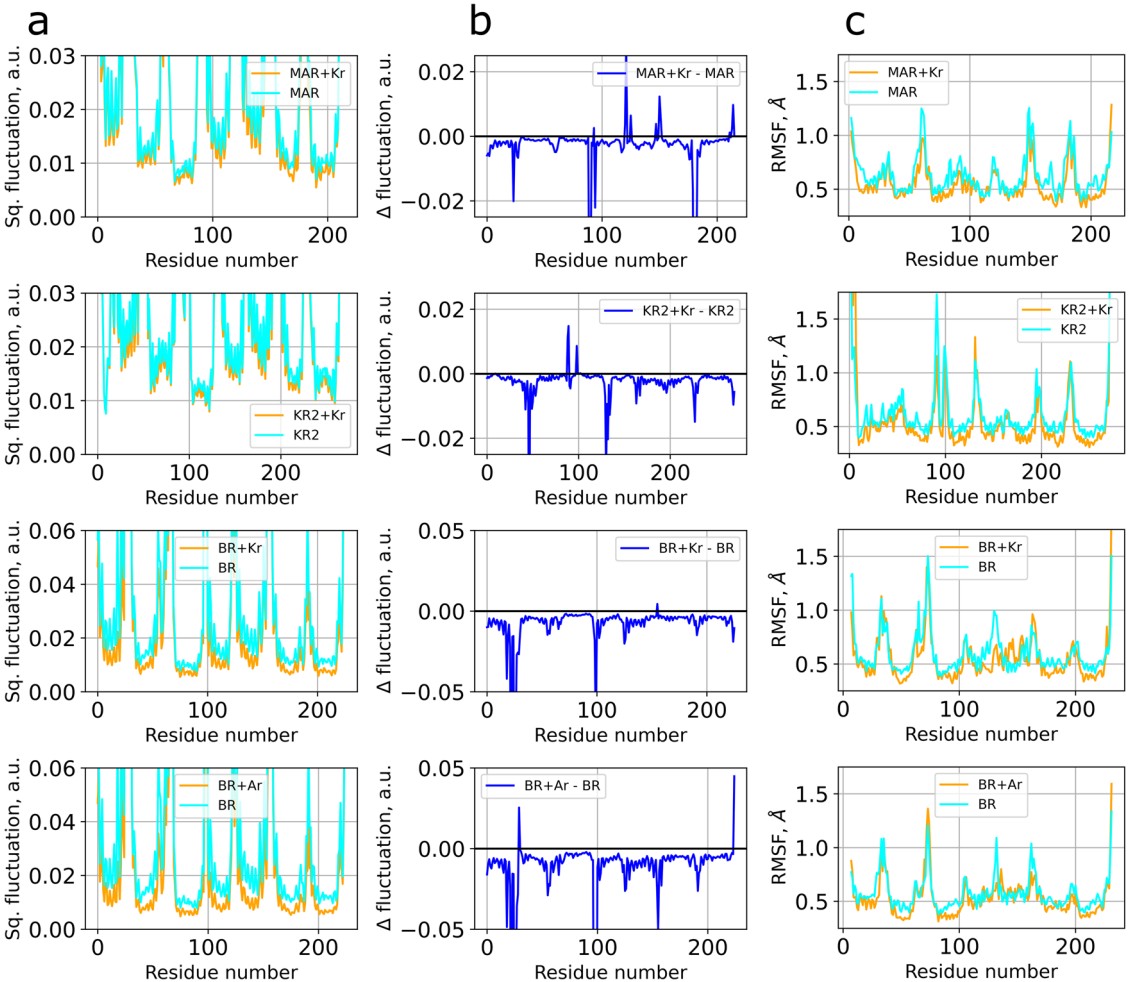

**Fig. 5 Squared fluctuations of amino acids calculated using anisotropic network model (ANM). a** Squared fluctuations for the four studied MP systems with and without bound noble gases (MAR, KR2, and tmBR (BR) with either krypton (Kr) or argon (Ar)). **b** Fluctuation differences (i.e., fluctuations in the presence of the noble gases minus fluctuations in their absence). **c** Per-residue root mean square fluctuations in the presence and in the absence of noble gases in a short all-atom MD simulation experiment.

(which can also explain the difference in anaesthetic potencies: neon and helium interact too weakly to produce narcotic effects). Besides, the process of noble gas binding in MPs constitutes therefore a good mimic of potential protein-lipid interactions. The lipid environment has been proved to be of a significant importance for proper functioning of MPs[52]. However, the studies providing a structure of a MP with systematically disturbed protein-lipid interactions across the entire MP surface are few. The striking result of this work is that noble gas atoms replace lipids at several binding sites on the surface of MPs, and our structural data show altered (by noble gas) protein-lipid interactions. Given that structural data on MPs crystallised *in meso* usually shows only fragmented electron densities of bound lipid chains, we suggest that lipid binding is not sufficiently strong along the whole length of the chain, and, therefore, hydrophobic molecules of smaller size (i.e. noble gases) may be competitive binders. However, this effect might be exaggerated by the substitution of native lipids with a crystallisation lipid medium in crystal structures. Indeed, native lipids should have a higher affinity to bind along the whole landscape of the hydrophobic surface of a MP, since they are more diverse and specific. In any case, one should not exclude that the effect caused by noble gas binding may be partially due to lipid displacement.

Our results from the MD also indicate that MP dynamics is suppressed in presence of a noble gas (Fig. 5, middle panels). We

suggest that this effect is significant and may impact transition of a MP between functional states similar to what has been previously demonstrated by Hayakawa and co-workers on wild-type bacteriorhodopsin (wtBR)[11]. Here, their studies showed that xenon atom binding to a pre-existing hydrophobic excavation on the surface formed between the C and D helices (located at the same depth in the membrane as Asp96, a key residue of proton uptake pathway) did not cause large structural rearrangements in the non-photolised state of wtBR. Nevertheless, the photocycle of purple membranes was remarkably modified: the decay of the M state was significantly accelerated, while the decay of the equilibrium states N and O was slowed down[11].

Binding to an internal cavity (see above and Supplementary Note 1) can clearly affect protein function by directly blocking substrate pathways. In the case of KR2 described here, surface-bound noble gases occupy excavations between α-helices allowing them to closely approach, and perhaps modify, the chemistry of essential co-factors (retinal) by modifying their environment (substantial structural perturbation of the end of helix E, see Fig. 3). Taking into account the absence of noble gas atoms in central cavities of KR2 molecule, it may be of interest to investigate the functional outcome.

However, the mechanism by which noble gas could trigger changes on the more general surface of a MP—their most common binding region as revealed by this work—is less evident. The

majority of binding sites we observe on the surface of the MPs studied are located in clefts between α-helices (see Fig. 1), sometimes these clefts may be 'locked' by nearby side chains and transformed into small cavities (see Supplementary Fig. 6). Such interactions are very similar to the allosteric modulation by anaesthetics that was recently observed for the GABA$_A$ receptor[35], nAChR[53] and GLIC-type receptor homologues[31,36]. The anaesthetic drugs showed a similar mechanism of action, embedding themselves in the voids between the subunits of a multimeric complex and stabilising their mutual positioning. Additionally, recent studies[54] showed a variety of allosteric binding sites on the surface of a GPCR. These facts therefore could be an indication that anaesthetic molecules rather act by locking conformational dynamics of a MP through binding to its membrane-buried hydrophobic surface.

These observations are consistent with our MD simulations, which indicate a reduction of MP dynamics upon noble gas atom binding (see Fig. 5, middle panels) and may also address the fundamental question as to why a number of structurally different anaesthetic compounds, including spherically symmetric noble gases, induce the same effect on a number of different targets. Indeed, this effect could be explained by non-specific binding to a MP at multiple allosteric sites at lower affinities compared to the direct effectors of the protein. This idea is further supported by previous work on the GABA$_A$ receptor[35], which showed that different molecules produce an anaesthetic effect while binding at different sites. A certain level of target selectivity is present though, with xenon, which does not affect GABA$_A$ receptors, being a good example. It is thus very unlikely to have a unified action site of anaesthetics as each substance acts on a wide range of proteins (producing different anaesthesia-related side effects)[24].

The study of nAChR[55] provides evidence that anaesthetics act merely by receptor desensitisation as opposed to non-anaesthetics that bind to the same allosteric site with similar affinities yet failing to prevent the ligand-induced conformational transition (nAChR pore opening). We therefore assume that suppression of protein dynamics is a common feature of the anaesthetics, considering our observations on MP dynamics suppression by noble gases and also the study on wtBR[11] in which xenon did not alter wtBR conformation while had an impact on the kinetics of the transition between different states in the photocycle. The dynamics-inhibition hypothesis is further supported by the reversal of anaesthesia with increasing ambient pressure[56], due to the fact that increasing the pressure enhances protein conformational dynamics[57]. We suggest that this observation (extensive interaction of noble gases with hydrophobic surface of MPs) may lead to a common idea of how anaesthetics work in general, which might be through the inhibition of transitional dynamics between states. However, further research needs to be undertaken in this direction.

## Methods

**Protein expression and purification**. Expression and purification of the sodium-pumping rhodopsin KR2 (KR2, UniProt ID N0DKS8) and proton-pumping rhodopsin MAR (MAR, UniProt ID S5DM51) were performed as described in ([58]) and ([59]), respectively.

**Expression and purification of bacteriorhodopsin with three mutations**. We used the native halobacterial nucleotide sequence with two silent mutations (C9A, G12A corresponding to amino acids Ala2 and Gln3) as reported before[60] to express BR in *E. coli*. We appended to the native sequence of bacteriorhodopsin (BR, UniProt ID P02945) at the 3′ terminus additional GSGIEGRSGAPHHHHHHHH* tag, which was used for metal-affinity chromatography purification and contained FXa cleavage site, and introduced the gene into the pSCodon 2.1 expression vector (Delphi Genetics) via *Nde*I and *Xho*I restriction sites. The mutations T17A, T24A, and T47A were introduced by PCR. The protein was expressed and purified as described[60] with slight modifications. *E. coli* BL21(DE3) cells were transformed

with pSCodon-derived plasmid and plated over LB-agar supplemented with ampicillin (0.05 mg/ml). The cells were grown at 37 °C in 2 L baffled flasks shaking at 120 rpm in an autoinducing medium ZYP-5052 containing ampicillin (0.05 mg/ml). At optical density OD$_{600}$ of 1.0 the temperature was decreased to 20 °C, and the media was supplemented with 10 μM all-*trans*-retinal and an additional portion of antibiotic. Another portion of antibiotic was added at OD$_{600}$ = 3.0. After 14 h of cultivation, the cells were harvested by centrifugation at 5000 rpm, resuspended in 20 mM Tris-HCl pH 8.0 supplemented with lysozyme (0.2 mg/ml) and DNAseI (0.02 mg/ml), and disrupted in M-110P Lab Homogenizer (Microfluidics, USA). Then 5 M NaCl was added to a final concentration of 200 mM and the suspension was layered over a glycerol cushion (3 ml—90% w/v) in 32 ml tubes. The total membranes were isolated by ultracentrifugation in SW-32Ti rotor (Beckman Coulter, Krefeld, Germany) at 28 000 rpm for 1 h, resuspended them in 20 mM Tris-HCl pH 8.0, 100 mM NaCl and solubilised overnight in DDM. The insoluble fraction was removed by ultracentrifugation (90,000 *g*, 1 h, 4 °C) and 10 mM of imidazole was added to the supernatant. Then it was loaded on a Ni-NTA column (Qiagen, Germany) and after washing the column with 5 volumes of 50 mM NaH$_2$PO$_4$/Na$_2$HPO$_4$ pH 8.0, 100 mM NaCl, 50 mM imidazole, 0.2% DDM and 1 volume of 50 mM NaH$_2$PO$_4$/Na$_2$HPO$_4$ pH 6.0, 100 mM NaCl, 50 mM imidazole, 0.2% DDM buffers we eluted the protein in a buffer containing 50 mM NaH$_2$PO$_4$/Na$_2$HPO$_4$ pH 6.0, 100 mM NaCl, 0.5 M imidazole and 0.3% DDM. Eluted protein applied to 24 ml Superdex 200i (GE Healthcare, Germany) column equilibrated with 50 mM NaH$_2$PO$_4$/Na$_2$HPO$_4$ pH 6.0, 100 mM NaCl, 0.2% DDM, and pooled a peak of coloured functional protein. Protein-containing coloured fractions were collected and concentrated to 60 mg/ml for crystallisation.

**Protein crystallisation**. All crystals were grown using the *in meso* approach[61] used in our previous works[62]. MAR, tmBR, and KR2 in their respective buffers (see above) were added to the monoolein-formed lipidic phase (Nu-Chek Prep, USA). The protein-LCP (lubic lipidic phase) mixtures (100-nl aliquots for each protein) were spotted on 96-well LCP glass sandwich plates (Marienfeld, Germany) and covered with 800 nl of precipitant solution using the NT8-LCP crystallisation robot (Formulatrix, USA). Crystals were grown at 20 °C and reached their final sizes within 1 to 8 weeks.

The best crystals of tmBR were obtained using 3.0 M ammonium sulfate and 0.1 M sodium acetate (pH 4.6) (Qiagen, Germany) as the precipitant solution and grew up to 50-100 μm in size. For KR2, the best crystals were obtained using 2.0 M sodium malonate (pH 4.3) (Hampton Research, USA) as a precipitant solution and grew up to 70-100 μm in size. The best crystals of MAR were obtained using 2.6 M ammonium sulfate and 0.1 M sodium acetate pH 5.2 (Qiagen, Germany) as the precipitant solution and were 100 to 150 μm in size.

Once crystals reached their final size, crystallisation wells were opened as described elsewhere[46]. Drops containing the protein-mesophase mixture were covered with 50 μl of the respective precipitant solution. Native crystals were loaded onto MicroMounts (MiTeGen, USA), flash-cooled, and stored in liquid nitrogen.

**"Soak-and-freeze" method for protein crystal derivatisation with noble gases**. At room temperature, noble gas atoms diffuse rapidly in and out of crystals, and might be completely desorbed after pressure release, in particular for argon and krypton that have small van der Waals radius. Hence, to produce proper derivatives, protein crystals require to be frozen while they are still under pressure[47,48]. The thermodynamic principle, the scheme and the photo of the device of the "soak-and-freeze" method that was designed to produce noble gas derivatives this way is described in a previous work[47]. The crystal was mounted with a cryoprotecting mother liquor inside a capillary on a specific pin designed for high-pressure experiments (SPINE compatible pin, MiTeGen LLC, Ithaca, New York, USA). The pressure cell consisted of a vertically oriented tube with its upper part maintained at room temperature and connected with the pressurising system, while its closed bottom part was regulated at cryogenic temperatures (~125 K). The pin with the sample was placed and held by a magnet at the top side of the cell at ambient conditions. Then, the tube was filled and pressurised with the relevant gas (at 150 bar and 2000 bar for krypton and argon, respectively) by the pressurising device. Then, after 10 min of pressurising/soaking time at ambient temperature, the crystal was flash-frozen by dropping the pin into the cold gas phase at the bottom of the cell, which was at cryogenic temperature. Finally, the pressure was released and the pin (with the frozen crystal) was transferred and recovered from the tube into liquid nitrogen, where subsequently it was mounted on a magnetic base, covered by a vial and stored in pucks for further data collection.

**Crystallographic data collection, structure solution, and refinement**. X-ray diffraction data were collected at the European Synchrotron Radiation Facility (ESRF), on beamlines ID23-1[63] and ID29[64] (Table 1). Prior to data collection, crystals mounted on the goniometer were located and characterised using X-ray mesh scans analysed by Dozor-MeshBest[65,66]. The X-ray beam size at sample position was chosen in the range from 10 to 50 μm FWHM in accordance with Dozor-MeshBest analysis. The experimental parameters for optimal data collection were designed using the program BEST[67]. Structure factor amplitudes and anomalous differences of krypton derivatives were measured at 14.3-14.4 keV

(f''=3.8$e$, Table 1), above the krypton K-absorption edge (which is 14.3256 keV, Table 1); the anomalous difference data of tmBR-Ar derivative were measured at lower energy, 6.7 keV ($f'' = 1.21e$); the two native datasets of MAR and one of tmBR were collected at the energy of 12.75 keV, at which another dataset of tmBR-Ar derivative was also collected to obtain higher resolution. For native crystals, standard cryogenic sample mounting was used. Each dataset was collected on a different single crystal. Some low-resolution datasets were discarded. Finally, the anomalous difference data were merged from three crystals of each of MAR-Kr and tmBR-Kr derivatives. One anomalous difference dataset was used for KR2-Kr. Two datasets were used for tmBR-Ar: one with anomalous difference data was used to identify argon positions and the other one of higher resolution was used to refine the structure.

Diffraction images were processed with XDS[68] and XSCALE[68]. Structure factor amplitudes and anomalous differences were generated using programs POINTLESS and AIMLESS[69]. Native KR2 structure (PDB code: 4XTL) was used as a starting model for KR2-Kr structure solution. Native MAR structure was refined based on the PDB code 5JSI. Native structure of tmBR was solved by molecular replacement using wild-type structure as starting model (PDB code 1QHJ). Final structural models (see Table 2) were obtained by alternating cycles of manual building in Coot[70] followed by refinement in REFMAC5[71]. Omit (F$_o$-F$_c$) maps were used to verify the presence of lipid fragments, ions, and water molecules initially found in the native structures. Anomalous difference Fourier maps were calculated with fast Fourier transform (FFT, CCP4) using DANO and ($\alpha_{calc}$ + 90°) as coefficients ($\alpha_{calc}$ = calculated phases from the final refined models). Noble gas atoms were placed in the coinciding peaks of the anomalous difference maps (above 5 r.m.s. level) and 2F$_o$–F$_c$ maps, their positions and occupancies were refined in REFMAC5 while keeping the B-factor close to that of nearest protein atoms. Figures were prepared using PyMOL (Schrödinger, LLC). Values r.m.s.d. for C$_\alpha$-atoms were calculated by PyMOL. Internal voids in protein structure were calculated by HOLLOW v1.3[72].

**Molecular dynamics simulations and normal mode analysis**. In each case, the simulation system consisted of a monomer model of tmBR, KR2, or MAR obtained in the present work. In all simulations, the proteins were embedded in a model bilayer consisting of DLPC for MAR and DPPC for all other systems and solvated with TIP3P water with counter ions by means of the CHARMM-GUI web-service[73]. DLPC for MAR was chosen to ensure better hydrophobic match between a somewhat hydrophobic region of MAR and DLPC membrane, which possesses shorter acyl chains. Then, a small fraction (approximately 2%) of water molecules were replaced by atoms of a specific noble gas resulting in their concentration of ~0.5–0.6 M. The excessive presence of noble gases allowed us to improve the sampling and, therefore, diminish possible drawbacks of limited simulation time[74–76]. On the other hand, the chosen concentration was low enough to prevent unwanted aggregation of noble gas atoms. The details of the simulated systems are provided in Table 3.

The recommended CHARMM-GUI protocols were followed for the initial energy minimisation and equilibration of the systems. The atoms of proteins and lipids in the systems are a subject for harmonic positional restraints. The steepest descent minimisation (5000 steps) was followed by a series of short equilibration simulations (25–50 ps) in the NPT ensemble using a Berendsen thermostat and barostat with the restraints on lipids gradually released. For the production simulations, a Nose–Hoover thermostat and Parrinello–Rahman barostat were used. In the production simulations, the C$_\alpha$-atoms of the protein backbone were constrained using the harmonic potential with a force constant of 100 kJ/mol/nm$^2$ to prevent large fluctuations of the protein structure, which could obscure further analysis. The temperature and pressure were set to 323.15 K and 1 bar with temperature and pressure coupling time constants of 1.0 ps$^{-1}$ and 0.5 ps$^{-1}$, respectively. All MD simulations were performed with GROMACS version 2020.2[77]. The time step of 2 fs was used for all production simulations. The CHARMM36 force field[78] was used for the proteins, lipids, and ions, while the parameters for the noble gases were obtained from Vrabeck *et al.*[79]. All production simulations were for 100 ns. The density maps for the noble gases were calculated using the *volmap* utility in VMD[80] after roto-translational alignment of a protein and the treatment of periodic boundary conditions. Only the last 90 ns of the production simulations were used for the analysis.

Density peaks (Table 4) were identified on the density maps sampled on a 0.5 Å grid, above {mean+5 r.m.s.d.} level as local maxima in a moving window of a size 2 Å. Density peaks coinciding with crystallographic noble gas sites within a distance of 4 Å were counted as "atom matches". Density peaks clusters (Table 4, Supplementary Figure 5) were identified by hierarchical cluster analysis using a cut-off distance of 5 Å between cluster centres. Clusters containing no "atom matches" were counted as "not present".

We also assessed the alteration in protein dynamics upon noble gas binding using the anisotropic network model (ANM) approach[81,82] based on application of the normal mode analysis (NMA) to a coarse-grained elastic network model (ENM). The latter included a single bead per each amino acid located at its C$_\alpha$ atom. All beads separated by less than 15 Å were considered connected by uniform harmonic potentials, i.e. "springs". Apart from that, noble gas resolved by the MX was also included into the elastic network and modelled as similar particles similarly to an ENM-based method for identification of "essential" binding sites,

ESSA[83]. We used the ProDy package to calculate normal modes and mean-square fluctuations of the coarse-grained beads estimating the fluctuations of atoms[84] for proteins with and without bound noble gases. Since the absolute values of fluctuations depend on the selection of the force constant in the harmonic potentials of the elastic network, with this latter being an arbitrary value, the resulting fluctuations are given in the arbitrary units.

The normal mode analysis was backed with short (10 ns each) all-atom simulations of all four protein systems with and without the crystallographically resolved argon and krypton atoms. The simulations were carried out using the same protocol as for the simulations with the excess of noble gases. The protein dynamics was estimated in terms of per-residue root mean square fluctuations.

Although ENM is a rather simple model, this method demonstrated surprisingly satisfactory results in many real applications, including prediction of dynamic regions[85,86], relevant conformational modes and collective motions[87,88], estimation of mechanical properties of biomolecules[89], etc.

**Reporting summary**. Further information on research design is available in the Nature Research Reporting Summary linked to this article.

## Data availability
All diffraction data and refined models have been deposited in the Protein Data Bank: 7Q38 (tmBR-argon), 7Q35 (tmBR-krypton), 7Q36 (KR2-krypton), 7Q37 (MAR-krypton). Any remaining information can be obtained from the corresponding author upon reasonable request.

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

## Acknowledgements

This study used the infrastructure of the Applied Genetics Resource Facility of MIPT (support grant 075-15-2021-684). The work was supported by the Ministry of Science and Higher Education of the Russian Federation No. 075-15-2021-1354 (X-ray data collection and treatment) and Russian Science Foundation No. 21-64-00018 (protein preparation). V.B. and I.O. acknowledge the Ministry of Science and Higher Education of the Russian Federation (agreement no. 075-00337-20-03, project FSMG -2020-0003).

## Author contributions

V.G., A.P. and P.C. designed the study; I.M. wrote the manuscript with the contribution of the other authors; M.R., K.K., R.A., D.B., Yu.R. and I.O. expressed and purified the proteins; T.B. supervised the expression and purification; M.R., K.K. and R.A. crystallised the proteins; P.C. and P.vdL. designed the high-pressure equipment; P.C. did high-pressure derivatisation of the crystals with a help from I.M. and M.R.; I.M. and A.P. collected the diffraction data; I.M. solved and refined the structures; G.B. and A.P. supervised diffraction data analysis; I.M. searched noble gas positions in the structures and did structure analysis with a help from G.B., S.B., K.K., C.M.D. and A.P.; P.O. did all MD calculations; P.O. and I.M. did analysis of the MD results; C.M.D., G.L., G.B., V.B. and P.C. helped with interpretation of the results of the study; I.M. prepared the figures with a help from P.O. and K.K.; C.M.D., A.P., V.G., K.K., P.O., G.B., P.vdL., P.C. and G.L. helped with manuscript preparation; P.C., A.P. and V.G. supervised the research.

## Competing interests

The authors declare no competing interests.
