## [Peer Review File · Communications Biology]

Reviewers' comments:

Reviewer #1 (Remarks to the Author):

It is known from long that noble gases interact with proteins and can be good anaesthetics, even if the rational is still not understood. As noble gases are known to interact preferentially with hydrophobic regions, authors solved structures of three membrane proteins by crystallography on crystals obtained in LCP (lipid cubic phase) in order to question the role of lipids. This is a novelty compared to the classic analysis restricted to the proteins' cavities. They submitted the crystals to pressurized argon or krypton before recording of anomalous data sets, as these gases have been proved to have anaesthetic effects when submitted to hyperbaric conditions. This crystallographic approach was completed by MD simulations performed both on the derived and native structures to analyse the probability density maps of gas positions. They obtain a good correlation on peaks observed on the surface of the proteins and they show that noble gas atoms are able to displace lipid molecules bound to protein surface. This could have consequence on the dynamic of the protein conformation changes that could be one of the possible explanations for the anaesthetic effect of noble gas.

I recommend major rewriting of the article before acceptance to clarify the conclusions but no additional experiments

Major point

The article is not very precise. All the phenomena are described roughly with no clear description of the interacting environment of the gas atoms.

Authors did a lot of work, recording a lot of data sets, solving six structures, analyzing them by MD simulation and NMA, but the exploitation of their results is quite limited.

They show some gas atoms modify some lipids positions but no description is performed of the amino-acid environment changes (or conservation).

They say that globally the peaks in the anomalous maps and in the MD simulations are localized at equivalent positions but, as an example, the interpretation of figure 4 has to be performed by the lector with very few accompaniment. This observation is not true for all the peaks but no reasons are given except for internal cavities.

Figure 4 "right side, delta fluctuation" is one of the most important result to justify authors conclusion, but here again it is not properly exploited. Why not to discuss about the regions of maximum difference, show the structures colored in function of these displacements and try to make a link with what is known of the protein function and its conformation changes. This would make the conclusion more convincing.

You do not explain why you studied these three particular proteins, that all present the same fold with almost no clear cavities, and why you used a mutated form of bacteriorhodopsin

Minor points

figure 1 legend, line 127: there is 2 "in the"

figure 2 legend, line 134: there is 2 "atoms"

figure 2 is too small to be able to see the displacement of the lipids. Be careful to make it big enough in the published version of the article

line 141: "fragments seen IN the native crystal structures"

line 161: I can't find the legends of the four videos but I have the sentiment authors have to refer to several ones here.

figure 4: right panel legends on images are too small. Maybe you can add the name of the protein/gas at the right of each line

line 198: "an in silico method" omit the "s"

line 207: omit "described above"

Methods

lines 308 and 310, what is the concentration in ampicillin?

idem line 312, precise the quantities of lysozyme and DNaseI used

Protein crystallisation

line 331, I can't find the respective crystallisation buffers...

line 333, do you mean "for each protein?"

line 377, indicate the K-absorption edge

lines 378-379, explain why you used different energies (λ , λ' ...)

the number of merged data sets is not clear, and even more difficult to understand when looking at table 1. Please try to clarify the text and make the table 1 consistent and explicit.

Table 2, occupancy: some noble gas positions present very low occupancies. How can you be sure it is noble gas and not water molecules or other ions? These atoms (water, ion) are not totally conserved between several data sets, especially at protein surface.

Table 4, lines "surface", "crystal" and "internal" atom matches: indicate the way of the ratio MX/MD

Reviewer #2 (Remarks to the Author):

I.

In the paper entitled 'High-pressure crystallography unravels noble gas intervention into protein-lipid interaction and suggests a model for anaesthetic action' the authors present a holistic approach to the problem of narcotic potency of the noble gases in hyperbaric environments. The anesthetic effects of two noble gases: krypton and argon that are well known as conditional anesthetics have been studied at the level of crystallographic structures of 4 transmembrane proteins (MP). The mentioned MPs have been exposed to a gaseous environment at standard conditions and further refined by high-pressure freezing method, at very high 130 bar (Kr) and 2000 bar (Ar) pressure and at the cryogenic temperature of 125 K. Within the limits of the proposed experiments, X-ray crystallography univocally points on the outer hydrophobic surface of the MPs as a one of the preferable binding sites for noble gases. This is also the preferable binding site for surrounding lipid molecules. Based on these findings, which are partly supported by all-atom (AA) classic molecular simulations experiments, authors conclude that lipid-MP proteins interactions could be at the focus of general anesthetic actions and might be the important factor determining affected MP functionality. The coarse-grained elastic network model (ENM), additionally deployed by the authors further elaborates the molecular model of inert gas narcosis (IGN) presenting it as a process presumably causing the slight decrease in conformational dynamics of MP in the presence of a noble gas.

II.

The paper is well written, the authors thoroughly explain the applied methodologies; data collection and subsequent data analysis. The paper definitely will be suitable for publication upon minor revisions. Prior to the submission of this paper, the authors have performed crystallographic and molecular studies and are experts in the field. This paper is therefore built on solid previously published experimental data, which have been however unable to clarify the complex modulation of MP activity by noble gases, paving the way for theoretical studies. Given the experience of the authors on this topic, the discussion of the results and the conclusions are properly presented. The specific revision points I suggest to consider are listed below.

Firstly, should be addressed the main concern raised by very high pressure applied on protein crystals exposed to the gas phase. The wide audience of the journal might not be familiar in detail with "Soak-and-freeze" method for protein crystal derivatization. The high pressure is a well-known thermodynamic factor reversing general anesthesia [54, Paper's references list]. Since it is unclear from the paper text how sustainable are the protein crystals and what is their conformational stability at high pressure, it would be worthy to deal with it at the discussion section. What if the molecular model the authors discuss dealing actually with high-pressure hyper-excitability of the central nervous system and not inert gas narcosis?

Secondly, regarding molecular simulations data - they might also suffer from the lack of clarity since the simulation boxes are too small; the explored proteins are presented as isolated transmembrane domains - TMD, without any visible extra-membrane part extended to the meaningful bulk solvent phase, which in turn can act as a competitive powerful thermodynamic 'sink', additionally to highly hydrophobic bilayer core. Moreover, the recently published simulation

works e.g.: 'Chen, J., Chen, L., Wang, Y. et al. Exploring the Effects on Lipid Bilayer Induced by Noble Gases via Molecular Dynamics Simulations. Sci Rep 5, 17235 (2015). <https://doi.org/10.1038/srep17235> ' show that non-immobilizer neon atoms also populate highly hydrophobic bilayer core, similarly with Ar and Kr, but not mandatory cause to anesthetic effect. I would expect some analysis and critique on such discrepancies.

III.

1. I would double check the claim at the line '61' - Introduction part, for validity of Meyer-Overton rule, given the present body of knowledge. Indeed, all general anesthetics (GA) comply with the rule, but not all the molecules that comply with the rule are general anesthetics. No common property between GA has been detected to date, although recent researches have pointed to, for instance, total dipole moment (TDM) of GA molecules, additionally to hydrophobicity. Please see the paper: 'QSAR analysis of compounds exhibiting general anesthetics' properties. L. Bober, P. Kawczak, T. Baczek. Letters in Drug Design & Discovery 9 (6), 595-603, 2012.' Large atoms of the anesthetic noble gases are mentioned at the paper.

2. The lines '71-72' - Introduction part. I would double check the pressure values mentioned. Ar and especially Kr are supposed to be more powerful anesthetics than nitrogen. The nitrogen narcotic effect, on the other hand, is endangering divers at pressure as low as 4 ATA.

3. Figure 4. Arrangement of peaks of the MD maps. Since the amino acid sequences are known for all proteins and presented on the maps, it would be interesting to see if there are any correlations between amino acids hydrophobicity index (or aromaticity?) and noble gas positions. PyMoL software also allows proteins surface visualization in terms of colorization of hydrophobic interactions. I suggest to the authors double-check this option. The white molecular surface has not much meaning in my opinion.

Reviewer #3 (Remarks to the Author):

Comments to the Author

The manuscript "High-pressure crystallography unravels noble gas intervention into protein-lipid interaction and suggests a model for anaesthetic action" by Melnikov et al. presents the crystal structures of three well-studied membrane proteins in the presence of noble gases, argon and krypton, the authors also present analysis of MD simulations of the same protein in the presence of noble gases. They have identified argon and krypton binding sites on these proteins and propose as a hypothesis that noble gases, by altering protein/lipid contacts, may affect protein function.

From my perspective, the results of this paper are quite robust, compelling, and of great interest. As a specialist in MD simulation, I cannot evaluate the methods used in the experimental part of the paper. But as far as the MD simulation part is concerned, I think that the analyses should have been deepened allowing to bring molecular details inaccessible to crystallography.

Concerns:

1) There are no control simulations (without gases). Since one of the assumptions of this paper is that noble gases, upon binding to the protein, displace the lipid in contact with the MP, one would expect the MD simulation to be able to document such an observation in more detail in a membrane bilayer and not in a crystal packing.

2) I'm worried by the length of simulation (100 ns). Given my experience with membrane protein simulation (Methods & Protocols 2017 Murail, https://doi.org/10.1007/978-1-4939-7151-0_20), membrane relaxation around the protein can take up to 20-100 ns, as ligand partition can reach equilibrium after more than 100 ns. That might also probably explain why the internal binding sites are not so much explored during the simulations.

3) The analysis of noble gas binding is only documented by an average density map. In my opinion, MD simulation has the great advantage over crystallographic studies in capturing the molecular details of such events.

One would expect to see more detailed information about the dynamics of noble gas binding. The reader has no idea of the stability of these interactions, the exchange rate of noble gases for example could be documented at a few binding sites. Similarly, the average residence time in each binding site or the average number of gas atoms in each pocket could be more informative than the density map alone.

4) I was not really convinced by the NMA results. If you could reproduce such a difference in dynamics with the protein in the presence of the crystallographic lipids and with or without Ar/Kr, I think that would make this result more robust.

Minor Concerns:

1) It was not clear to me whether the starting structures used in the MD simulations were from the presented work (native structures) or from previously published structures.

2) The choice of noble gas concentration/pressure should be discussed. This is a critical issue, as most gases are distributed in the membrane bilayer, and there is no clear answer to the question of what concentration or pressure you are mimicking in your simulation. You should indicate at least one concentration/pressure range that you are looking to mimic. In some of the simulated system the ratio Ar/Kr:lipid is reaching 2, that seems quite high. In supplementary information, I think it would be interesting for the reader to have a snapshot picture of each simulated systems at the end of the simulations, with lipid molecule and gas atoms.

3) DPPC/DLPC membrane composition is misleading. At first reading I thought it was a mixed membrane of DPPC:DLPC. Is there any reason for changing lipids composition for MAR only?

4) SI Figure 1. What does the red transparent surface represents ?

5) SI Figure 3. Not sure this figure is bringing any useful information.

Dear Reviewers, Editors,

We thank the reviewers for their remarks and comments, which helped us a lot to improve the manuscript. Below we respond to all of the comments point by point and address them in the main text. The newly added paragraphs we have highlighted in green in the revised version of the manuscript.

Yours sincerely,
Valentin Gordeliy and the co-authors

Response to Reviewers' comments:

Reviewer #1 (Remarks to the Author):

It is known from long that noble gases interact with proteins and can be good anaesthetics, even if the rational is still not understood. As noble gases are known to interact preferentially with hydrophobic regions, authors solved structures of three membrane proteins by crystallography on crystals obtained in LCP (lipid cubic phase) in order to question the role of lipids. This is a novelty compared to the classic analysis restricted to the proteins' cavities. They submitted the crystals to pressurized argon or krypton before recording of anomalous data sets, as these gases have been proved to have anaesthetic effects when submitted to hyperbaric conditions. This crystallographic approach was completed by MD simulations performed both on the derived and native structures to analyse the probability density maps of gas positions. They obtain a good correlation on peaks observed on the surface of the proteins and they show that noble gas atoms are able to displace lipid molecules bound to protein surface. This could have consequence on the dynamic of the protein conformation changes that could be one of the possible explanations for the anaesthetic effect of noble gas.

I recommend major rewriting of the article before acceptance to clarify the conclusions but no additional experiments

Major point

The article is not very precise. All the phenomena are described roughly with no clear description of the interacting environment of the gas atoms.

Authors did a lot of work, recording a lot of data sets, solving six structures, analyzing them by MD simulation and NMA, but the exploitation of their results is quite limited.

They show some gas atoms modify some lipids positions but no description is performed of the amino-acid environment changes (or conservation).

We thank the reviewer for this comment. Responding to the comment, we specify and analyse in more detail the interaction sites of noble gases in Results section (see lines 119-162, new text). We also point out (see lines 156-159, new text) that "overall the derivatised structures do not differ much from the native ones". We have extended this phrase adding that some side chain alteration occurs and supplemented this with Suppl. Figure 3 describing how a phenylalanine side chain was displaced upon krypton binding in MAR crystal structure.

Supplementary Figure 1. Structural changes in MAR upon binding of krypton atoms. The structure of MAR-Kr is coloured in blue, whereas the corresponding native structure (aligned to MAR-Kr structure by C α positions) is coloured in orange. The electron density map (2Fo-Fc) of the derivative data set is drawn in light blue at 1.2 r.m.s. level around residues of interest and clearly shows their alternative conformation (PHE22, GLU29, and a lipid fragment) relative to the native structure. Krypton atoms are shown as red spheres.

They say that globally the peaks in the anomalous maps and in the MD simulations are localized at equivalent positions but, as an example, the interpretation of figure 4 has to be performed by the lector with very few accompaniment. This observation is not true for all the peaks but no reasons are given except for internal cavities.

The main reason behind this discrepancy is that MD simulations were performed on a single molecule whereas X-ray structures are in fact crystalline, with in-membrane crystal contacts blocking some surface area for noble gas binding. Performing a separate MD experiment on a protein ensemble (2-4 molecules in a bilayer arranged as in a crystal), failed to see any accumulation of noble gases in the places of protein crystal contacts (data not shown). Presumably, one of the reasons for that is small observation time in the MD experiment – because noble gas atoms require some time to diffuse to area between molecules in crystal contact. All this we also point out in the text (see lines 182-185, 226-241, new text). Another issue is that crystal structures show an average position of a noble gas atom while in the MD experiment several positions nearby can be identified with high accuracy between which the atom may frequent presumably.

Figure 4 "right side, delta fluctuation" is one of the most important result to justify authors conclusion, but here again it is not properly exploited. Why not to discuss about the regions of maximum difference, show the structures colored in function of these displacements and try to make a link with what is known of the protein function and its conformation changes. This would make the conclusion more convincing.

The main idea this figure is carrying is that overall protein dynamics is suppressed – this is clearly observed in the rightmost panel where a constant negative difference in fluctuations is seen along the whole protein sequence. In fact, that shows the low level of specificity in noble gas (anaesthetic) action as dynamics suppression is uniform in the structure. Peaks in fluctuation difference come from flexible loops between α -helices – since these are more dynamic in general and the difference is larger. It is protein-specific and cannot be generalised. We have added a short paragraph to the revised text:

New text, lines 193-197: "A constant negative difference in fluctuations is seen along the whole protein sequence. In fact, that shows the low level of specificity in noble gas action as dynamics suppression is uniform in the structure. Peaks in fluctuation difference come from flexible loops between α -helices – since they are more dynamic in general and the difference is larger".

You do not explain why you studied these three particular proteins, that all present the same fold with almost no clear cavities, and why you used a mutated form of bacteriorhodopsin

Those proteins are rhodopsins. We needed membrane proteins which are well studied, can be well crystallized and routinely diffract to very high resolution. Indeed, the fold of these proteins is quite similar. Nevertheless, these proteins have different origin. Therefore, they are surrounded by different lipids in the membranes. Moreover, BR, in opposite to two other proteins, is originated from archaea. It is known that there is a principal difference between lipid content of bacterial and archaeal membranes. Therefore, we expect a general mechanism of noble gas interaction shared across all membrane proteins and also some specific differences due to specific features of hydrophobic surfaces of the proteins. Taking into account the comment of the reviewer, we have added a paragraph to the revised manuscript:

New text, lines 103-106: "... these rhodopsins, despite sharing a similar fold, are of different origins and, therefore, are naturally surrounded by different lipids (which makes the MPs quite different in the landscapes of hydrophobic surfaces)".

The mutated form of bacteriorhodopsin was used because it forms non-twinned crystals.

Minor points

figure 1 legend, line 127: there is 2 "in the"

figure 2 legend, line 134: there is 2 "atoms"

figure 2 is too small to be able to see the displacement of the lipids. Be careful to make it big enough in the published version of the article

line 141: "fragments seen IN the native crystal structures"

line 161: I can't find the legends of the four videos but I have the sentiment authors have to refer to several ones here.

figure 4: right panel legends on images are too small. Maybe you can add the name of the protein/gas at the right of each line

line 198: "an in silico method" omit the "s"

line 207: omit "described above"

We are sorry for these typos and thank the reviewer very much for careful reading of the manuscript. We have corrected the text correspondingly.

Methods

lines 308 and 310, what is the concentration in ampicillin?

idem line 312, precise the quantities of lysozyme and DNaseI used

We agree that the values of the concentrations should be specified. The concentrations have been added (see lines 337-343, new text).

Protein crystallisation

line 331, I can't find the respective crystallisation buffers...

Those are protein buffers (after purification); the crystallisation buffers were described further in the text:

Lines 367-372, new text: "The best crystals of tmBR were obtained using 3.0 M ammonium sulfate and 0.1 M sodium acetate (pH 4.6) (Qiagen, Germany) as the precipitant solution and grew up to 50-100 μ m in size. For KR2, the best crystals were obtained using 2.0 M sodium malonate (pH 4.3) (Hampton Research, USA) as a precipitant solution and grew up to 70-100 μ m in size. The best crystals of MAR were obtained using 2.6 M ammonium sulfate and 0.1 M sodium acetate pH 5.2 (Qiagen, Germany) as the precipitant and were 100 to 150 μ m in size."

line 333, do you mean "for each protein?"

line 377, indicate the K-absorption edge

lines 378-379, explain why you used different energies (λ , λ' ...)

the number of merged data sets is not clear, and even more difficult to understand when looking at table 1. Please try to clarify the text and make the table 1 consistent and explicit.

We have added the clarifications to the revised text (see lines 406-416, new text).

Table 2, occupancy: some noble gas positions present very low occupancies. How can you be sure it is noble gas and not water molecules or other ions? These atoms (water, ion) are not totally conserved between several data sets, especially at protein surface.

Using anomalous difference maps provides a very strong indication of noble gas atom presence, as anomalous signal cannot come from water. We indeed see anomalous signal from sulphur (methionine and SO_4^{2-}), however, the general sense has also allowed discriminating between noble gas and ions (i.e. ions cannot be found in depth of the membrane).

Table 4, lines "surface", "crystal" and "internal" atom matches: indicate the way of the ratio MX/MD

It has been corrected.

We thank again the reviewer very much for the valuable remarks.

Reviewer #2 (Remarks to the Author):

I.

In the paper entitled 'High-pressure crystallography unravels noble gas intervention into protein-lipid interaction and suggests a model for anaesthetic action' the authors present a holistic approach to the problem of narcotic potency of the noble gases in hyperbaric environments. The anesthetic effects of two noble gases: krypton and argon that are well known as conditional anesthetics have been studied at the level of crystallographic structures of 4 transmembrane proteins (MP). The mentioned MPs have been exposed to a gaseous environment at standard conditions and further refined by high-pressure freezing method, at very high 130 bar (Kr) and 2000 bar (Ar) pressure and at the cryogenic temperature of 125 K. Within the limits of the proposed experiments, X-ray crystallography univocally points on the outer hydrophobic surface of the MPs as a one of the preferable binding sites for noble gases. This is also the preferable binding site for surrounding lipid molecules. Based on these findings, which are partly supported by all-atom (AA) classic molecular simulations experiments, authors conclude that lipid-MP proteins interactions could be at the focus of general anesthetic actions and might be the important factor determining affected MP functionality. The coarse-grained elastic network model (ENM), additionally deployed by the authors further elaborates the molecular model of inert gas narcosis (IGN) presenting it as a process presumably causing the slight decrease in conformational dynamics of MP in the presence of a noble gas.

II.

The paper is well written, the authors thoroughly explain the applied methodologies; data collection and subsequent data analysis. The paper definitely will be suitable for publication upon minor revisions. Prior to the submission of this paper, the authors have performed crystallographic and molecular studies and are experts in the field. This paper is therefore built on solid previously published experimental data, which have been however unable to clarify the complex modulation of MP activity by noble gases, paving the way for theoretical studies. Given the experience of the authors on this topic, the discussion of the results and the conclusions are properly presented. The specific revision points I suggest to consider are listed below.

Firstly, should be addressed the main concern raised by very high pressure applied on protein crystals exposed to the gas phase. The wide audience of the journal might not be familiar in detail with "Soak-and-freeze" method for protein crystal derivatization. The high pressure is a well-known thermodynamic factor reversing general anesthesia [54, Paper's references list]. Since it is unclear from the paper text how sustainable are the protein crystals and what is their conformational stability at high pressure, it would be worthy to deal with it at the discussion section. What if the molecular model the authors discuss dealing actually with high-pressure hyper-excitability of the central nervous system and not inert gas narcosis?

We agree that at such pressures (130bar, 2000bar) it may be possible that the anaesthetic effect is reversed and the hyper-excitability effect takes place. However, considering that hyper-excitability of the central nervous system is the pressure effect per se, we assume that the same general principle of noble gas binding is valid also for anaesthesia-relevant pressures. Indeed, as noble gas occupancy at each binding site monotonously depends on the pressure we conclude that the binding is still present at anaesthesia-relevant pressures. The only purpose of high pressure here was in a thermodynamic way to populate binding sites across proteins to clearly see the overall picture with the means of X-ray crystallography.

Secondly, regarding molecular simulations data - they might also suffer from the lack of clarity since the simulation boxes are too small; the explored proteins are presented as isolated transmembrane domains – TMD, without any visible extra-membrane part extended to the meaningful bulk solvent phase, which in turn can act as a competitive powerful thermodynamic 'sink', additionally to highly hydrophobic bilayer core. Moreover, the recently published simulation works e.g.: 'Chen, J., Chen, L., Wang, Y. et al. Exploring the Effects on Lipid Bilayer Induced by Noble Gases via Molecular Dynamics Simulations. Sci Rep 5, 17235 (2015). <https://doi.org/10.1038/srep17235> ' show that non-immobilizer neon atoms also populate highly hydrophobic bilayer core, similarly with Ar and Kr, but not mandatory cause to anesthetic effect. I would expect some analysis and critique on such discrepancies.

We understand and appreciate the comment of the reviewer. For the MD simulations we used full sequences of the proteins, not just TMD; the proteins do have an interface with bulk solvent phase (represented by simulated water molecules and ions), though they do not have a separate cytoplasmic domain. Taking a protein model that has a large soluble domain would not be beneficial in our opinion as it would show additionally only noble gas interaction with soluble protein which is studied elsewhere and is not of concern in our work.

As polarisability is the main term in London force xenon has therefore the strongest interaction. With lighter atoms: krypton, argon, neon and helium – their polarisability drops down rapidly and their energetic effect on a MP structure is less pronounced per mol of noble gas. For that reason, they require higher pressure (i.e. by the law of mass) that xenon to produce energetic effect of the same scale. Even though neon and helium tend to concentrate in the bilayer, they are just too weak to produce anaesthetic effect at higher pressure up to the value where anaesthetic reversal takes place and where hyper excitability and convulsions start). Taking into account the reviewer remark, we comment this point in the text:

New text ,lines 75-77: "presumably because these require pressure values far beyond the limits of a nervous system, at which the hyper excitability already takes place"

III.

1. I would double check the claim at the line '61' - Introduction part, for validity of Meyer-Overton rule, given the present body of knowledge. Indeed, all general anesthetics (GA) comply with the rule, but not all the molecules that comply with the rule are general anesthetics. No common property between GA has been detected to date, although recent researches have pointed to, for instance, total dipole moment (TDM) of GA molecules, additionally to hydrophobicity. Please see the paper: 'QSAR analysis of compounds exhibiting general anesthetics' properties. L. Bober, P. Kawczak, T. Baczek. Letters in Drug Design & Discovery 9 (6), 595-603, 2012.' Large atoms of the anesthetic noble gases are mentioned at the paper.

We agree with the comment. We took into account the reviewer critics and modified the corresponding part of the introduction:

New text, lines 60-63: "A protein alone as the target, however, does not explain why anaesthetic potency of a substance varies in conformity with its solubility in lipids (the Meyer-Overton rule). It would be therefore evident to focus on membrane proteins (MPs)."

2. The lines '71-72' – Introduction part. I would double check the pressure values mentioned. Ar and especially Kr are supposed to be more powerful anesthetics than nitrogen. The nitrogen narcotic effect, on the other hand, is endangering divers at pressure as low as 4 ATA.

This is a correct remark, the values MAC were given for rats, we have noted it now in the revised manuscript (line 73, new text).

3. Figure 4. Arrangement of peaks of the MD maps. Since the amino acid sequences are known for all proteins and presented on the maps, it would be interesting to see if there are any correlations between amino acids hydrophobicity index (or aromaticity?) and noble gas positions. PyMol software also allows proteins surface visualization in terms of colorization of hydrophobic interactions. I suggest to the authors double-check this option. The white molecular surface has not much meaning in my opinion.

Thank you very much for the comment. We have added Suppl. Figure 2 showing a histogram of amino acid frequency in the noble gas 5Å-environment. The histogram is also supplemented with amino acid hydrophobicity scale at the background. We, nevertheless, think that surface colourisation would complicate reading any of the figures; it is clear as so that the membrane-exposed area of a MP is highly hydrophobic.

Supplementary Figure 2. Frequency of each residue in a noble gas atom environment. The value of frequency was calculated as the number of atoms of each particular residue in a 5Å-vicinity of noble gas atoms, normalised to the frequency of natural occurrence of this residue. An empirical hydrophobicity scale is shown at the background and depicts free energy ΔG required to put a residue into the core of lipid bilayer (right scale).

We thank again the reviewer very much for the valuable remarks.

Reviewer #3 (Remarks to the Author):

Comments to the Author

The manuscript "High-pressure crystallography unravels noble gas intervention into protein-lipid interaction and suggests a model for anaesthetic action" by Melnikov et al. presents the crystal structures of three well-studied membrane proteins in the presence of noble gases, argon and krypton, the authors also present analysis of MD simulations of the same protein in the presence of noble gases. They have identified argon and krypton binding sites on these proteins and propose as a hypothesis that noble gases, by altering protein/lipid contacts, may affect protein function.

From my perspective, the results of this paper are quite robust, compelling, and of great interest. As a specialist in MD simulation, I cannot evaluate the methods used in the experimental part of the paper. But as far as the MD simulation part is concerned, I think that the analyses should have been deepened allowing to bring molecular details inaccessible to crystallography.

We thank the reviewer for taking time for assessing our work and for providing the valuable comments. We address to our best the comments of the reviewer; we have provided the necessary analysis, included several additional figures, and changed the text where appropriate.

Concerns:

1) There are no control simulations (without gases). Since one of the assumptions of this paper is that noble gases, upon binding to the protein, displace the lipid in contact with the MP, one would expect the MD simulation to be able to document such an observation in more detail in a membrane bilayer and not in a crystal packing.

Limited computational power does not allow us to significantly extend the simulation time of existing calculations, as well as to carry out additional simulations. However, we tried to carry out more accurate analysis of available trajectories, including the analysis of the binding kinetics of inert gases (see our answers to the following issues).

2) I'm am worried by the length of simulation (100 ns). Given my experience with membrane protein simulation (Methods & Protocols 2017 Murail, https://doi.org/10.1007/978-1-4939-7151-0_20), membrane relaxation around the protein can take up to 20-100 ns, as ligand partition can reach equilibrium after more than 100 ns. That might also probably explain why the internal binding sites are not so much explored during the simulations.

The reviewer is absolutely right that the convergence of MD trajectories and sufficient sampling is an important problem. Indeed, the slowest degrees of freedom in lipids need up to 100 ns to equilibrate [10.1021/acs.jctc.5b01106] while for membrane proteins even longer simulations are often required [10.1002/prot.21308]. However, since our major focus was not to thoroughly sample lipid binding itself but rather to explore the binding of inert gases and, due to limited computing resources, we were bound by the relatively short simulation time in the present study. We tried to compensate for the potential sampling problem by performing simulations in the presence of excess noble gases. The motivation behind this approach and the chosen gas concentrations are discussed below (answer to minor concern #2). We added the corresponding text with a caution regarding the incomplete sampling due to the short simulations:

New text, lines 441-444: "The excessive presence of noble gases allowed us to improve the sampling and, therefore, diminish possible drawbacks of limited simulation time [10.1021/acs.jctc.5b01106; 10.1002/prot.21308; 10.1007/978-1-4939-7151-0_20]. On the other hand, the chosen concentration was low enough to prevent unwanted aggregation of noble gases."

3) The analysis of noble gas binding is only documented by an average density map. In my opinion, MD simulation has the great advantage over crystallographic studies in capturing the molecular details of such events.

One would expect to see more detailed information about the dynamics of noble gas binding. The reader has no idea of the stability of these interactions, the exchange rate of noble gases for example could be documented at a few binding sites. Similarly, the average residence time in each binding site or the average number of gas atoms in each pocket could be more informative than the density map alone.

We thank the reviewer for these valuable suggestions. In fact, the detailed analysis of binding affinities of noble gases performed in the revised manuscript has shown that the observed interactions are rather strong with the binding free energy estimated (using the Boltzmann equation $dG = -RT \ln(p/p_0)$, where p_0 and p are

the gas densities in bulk and in the binding site) down to -16 – -19 kJ/mol (i.e. up to almost 8 kT). Such affinity precludes very fast exchange of bound gas atoms especially in the buried sites. We have included an additional figure in the SI (Suppl. Figure 5) illustrating kinetics of the gas binding in several representative sites. The corresponding text and the figure were added to the manuscript:

New text, lines 174-178: “The binding free energy corresponding to these peaks was estimated down to -16 – -19 kJ/mol (i.e. almost 8 kT at 323 K, see Table 3) using the Boltzmann relation as suggested in [10.1021/ct300117j]. Interestingly, the inspection of individual binding events at some representative sites revealed that noble gases form considerably long-lived contacts lasting for 10-100 ns (see figure S1).”

Suppl. Figure 5. Analysis of binding events at representative sites in 4 simulated systems. Black dots indicate that the corresponding gas atom is bound at the site. (a-b) KR2+Kr; (c-d) MAR+Kr; (e-f) BR+Ar; (g-h) BR+Kr.

4) I was not really convinced by the NMA results. If you could reproduce such a difference in dynamics with the protein in the presence of the crystallographic lipids and with or without Ar/Kr, I think that would make this result more robust.

We thank the reviewer for pointing out this important issue. While the ENM (elastic network model) variant of normal mode analysis (NMA) utilized in our present work is indeed based on rather simple assumptions and, strictly speaking, is only valid for prediction of protein dynamics in the vicinity of the potential energy minimum, the method demonstrates surprisingly satisfactory results in many real applications, including prediction of dynamic regions [DOI: 10.1021/cr900095e, 10.1146/annurev.biophys.093008.131258], relevant conformational modes and collective motions [DOI: 10.1073/pnas.0904214106, 10.1111/php.12763], estimation of mechanical properties [DOI: 10.1371/journal.pcbi.1007327], etc. Particularly, an ENM-based method, Essential Site Scanning Analysis (ESSA), was proposed for identification of “essential” sites, i.e. residues that would significantly alter the protein’s global dynamics if bound to a ligand [DOI: 10.1016/j.csbj.2020.06.020]. The authors chose to mimic the crowding induced upon ligand binding by adding additional nodes to the ENM, and they used the resulting shifts in soft mode frequencies as a metric for evaluating the essentiality of each residue. The method, somewhat reminiscent of the one applied here by us, allowed the authors to identify both allosteric and orthosteric binding sites in a broad range of proteins.

Still, we preferred to back the present data with all-atom simulations of all four protein systems with and without the crystallographically resolved Ar/Kr atoms. Overall, the protein dynamics estimated in terms of per-residue root mean square fluctuations show similar trends as in the case of NMA-based estimation. The presence of noble gases restricts the dynamics of all examined proteins.

We added appropriate changes to the text and Figure 4 (splitting it into new Figure 4 and Figure 5):

New text, lines 197-201: “In order to cross-validate the results of NMA, we also conducted a series of short MD simulations with and without the crystallographically resolved atoms of noble gases. The results of these simulations appear in agreement with NMA indicating slightly reduced dynamics of proteins in the presence of Kr/Ar (see Figure 5).”

New text, lines 481-489: “The normal mode analysis was backed with short (10 ns each) all-atom simulations of all four protein systems with and without the crystallographically resolved Ar/Kr atoms. The simulations were carried out using the same protocol as for the simulations with the excess of noble gases. The protein dynamics estimated in terms of per-residue root mean square fluctuations. Although ENM is a rather simple model, this method demonstrated surprisingly satisfactory results in many real applications, including prediction of dynamic regions [DOI: 10.1021/cr900095e, 10.1146/annurev.biophys.093008.131258], relevant conformational modes and collective motions [DOI: 10.1073/pnas.0904214106, 10.1111/php.12763], estimation of mechanical properties of biomolecules [DOI: 10.1371/journal.pcbi.1007327], etc.”

Minor Concerns:

1) It was not clear to me whether the starting structures used in the MD simulations were from the presented work (native structures) or from previously published structures.

We apologize that the text was not clearly stating that the simulations were run for the structures obtained in the present work. We have modified the text accordingly:

New text, lines 434-435: "... monomeric model of tmBR, KR2, or MAR obtained in the present work."

2) The choice of noble gas concentration/pressure should be discussed. This is a critical issue, as most gases are distributed in the membrane bilayer, and there is no clear answer to the question of what concentration or pressure you are mimicking in your simulation. You should indicate at least one concentration/pressure range that you are looking to mimic. In some of the simulated system the ratio Ar/Kr:lipid is reaching 2, that seems quite high.

We thank the reviewer for asking this important question, which was not properly addressed in the manuscript. In order to improve the sampling of Ar/Kr binding poses within the limited simulation time (see the answer to major concern #2 above), we decided to simulate the protein systems at rather high concentration of probes (i.e. Ar/Kr atoms), i.e. 0.5-0.6 M. The ultimate concentration range was dictated by successful "druggability" simulations [DOI: 10.1021/ct300117j], where a similar approach although with different types of probe molecules was used to explore potential binding sites of drug targets. Typical concentrations of probes in such simulations are in the range of 0.1-1 M. The reason for this choice is two-fold: (a) improve sampling; (b) prevent aggregation of probes. In this work, we followed the same logic and opted for a concentration of 0.5-0.6 M. The corresponding text has been added to the manuscript:

New text, lines 170-173: "The MD simulations were carried out on monomeric molecular models of tmBR, KR2, and MAR in the presence of excessive concentration of noble gases (see Table 3) sufficiently high to improve the sampling but at the same time causing no aggregation."

New text, lines 440-441: "Then, a fraction (approx. 2%) of water molecules was replaced by atoms of specific noble gases resulting in their concentration of ~0.5-0.6 M."

In supplementary information, I think it would be interesting for the reader to have a snapshot picture of each simulated systems at the end of the simulations, with lipid molecule and gas atoms.

We have added the representative snapshots of the systems (see Suppl. Figure 7).

Suppl. Figure 7. Representative snapshots of the initial (each panel, left) and final (each panel, right) configurations of the simulated systems: MAC (a), KR2 (b), tmBR with krypton (c), and tmBR with argon (d). The protein molecules are shown using cartoon representation coloured accordingly to the secondary structure; noble gas atoms are shown as yellow spheres; phosphorus atoms of the membrane are shown as red spheres. Water, ions, and all lipid atoms except phosphorus are omitted for clarity.

3) DPPC/DLPC membrane composition is misleading. At first reading I thought it was a mixed membrane of DPPC:DLPC. Is there any reason for changing lipids composition for MAR only?

We are sorry about this misleading statement. DLPC was chosen for MAR system only for better hydrophobic match between somewhat shorter hydrophobic region of this protein and DLPC membrane, which possesses shorter acyl chains. We have fixed the corresponding text and explained the reasoning behind choosing DLPC lipid for MAR:

New text, lines 436-439: "... consisting of DLPC for MAR and DPPC for all other systems. The former was chosen to ensure better hydrophobic match between a somewhat shorter hydrophobic region of MAR and DLPC membrane, which possesses shorter acyl chains."

4) SI Figure 1. What does the red transparent surface represents?

It represents internal voids in the protein structures. We have added a mention to that in the figure caption and in Methods section:

New text, line 431: "Internal voids in protein structure were calculated by HOLLOW v1.3".

5) SI Figure 3. Not sure this figure is bringing any useful information.

This figure has been removed.

We thank again the reviewer very much for the valuable remarks.

REVIEWERS' COMMENTS:

Reviewer #1 (Remarks to the Author):

The authors answered all of my points and I feel the manuscript is clearer in this form. I particularly appreciate the addition of figure supp 2 that corresponds to my expectations. The manuscript can be published in this form

Reviewer #2 (Remarks to the Author):

The authors thoroughly considered all of the reviewers' suggestions in their revised version and considerably improved the manuscript's quality. The data presented by the authors may pave the way for more complex systems affected by general anesthetics, such as lipids, transmembrane proteins, and cytoplasmatic domains, to be studied further. The high pressure crystallography coupled with molecular simulations emphasize molecular mechanisms by which anesthetic noble gases might deprive signal transduction in membrane proteins. In light of the above, I recommend that the manuscript be published.

Reviewer #3 (Remarks to the Author):

The new version of the manuscript "High-pressure crystallography unravels noble gas intervention into protein-lipid interaction and suggests a model for anaesthetic action" by Melnikov et al. has been improved.

Most of my concerns have been addressed. I am still concerned about the length of the simulations, but I understand the lack of computational time. However, I think the impressive amount of work presented in this paper will make it a landmark paper on the molecular action of noble gases on membrane proteins.

Regarding the paper I mentioned (Methods & Protocols 2017 Murail, https://doi.org/10.1007/978-1-4939-7151-0_20), I didn't mean to suggest you cite it. Feel free to delete the citation if you don't find it appropriate.